# On people's perceptions of climate change and its impacts in a hotspot of global warming

**Parbati Phuyal**[1,2]☯*, **Isabelle Marie Kramer**[1,3]☯, **Indira Kadel**[4], **Edwin Wouters**[5,6], **Axel Magdeburg**[1], **David A. Groneberg**[1], **Ulrich Kuch**[1], **Bodo Ahrens**[7], **Mandira Lamichhane Dhimal**[1,8], **Meghnath Dhimal**[1,9]‡, **Ruth Müller**[1,3,10]‡

1 Institute of Occupational Medicine, Social Medicine and Environmental Medicine, Goethe University, Frankfurt am Main, Germany, 2 Institute of Environment and Sustainable Development, University of Antwerp, Antwerp, Belgium, 3 Department of Biomedical Science, Faculty of Pharmaceutical, Biomedical and Veterinary Sciences, University of Antwerp, Antwerp, Belgium, 4 Department of Hydrology and Meteorology, Kathmandu, Nepal, 5 Centre for Population, Family & Health, University of Antwerp, Antwerp, Belgium, 6 Centre for Health Systems Research & Development, University of the Free State, Bloemfontein, South Africa, 7 Institute for Atmospheric and Environmental Sciences, Goethe University, Frankfurt am Main, Germany, 8 Planetary Health Research Centre, Kathmandu, Nepal, 9 Nepal Health Research Council, Kathmandu, Nepal, 10 Unit Entomology, Institute of Tropical Medicine, Antwerp, Belgium

☯ These authors contributed equally to this work.
‡ MD and RM share first authorship on this work.
* phuyalparbati@gmail.com

**Data Availability Statement:** All relevant data are within the paper and its Supporting Information files.

## Abstract

The Hindu Kush Himalayan region is a global hotspot for climate change and highly vulnerable to its direct and indirect impacts. Understanding people's perception of climate change is crucial for effective adaptation strategies. We conducted a study by using quantitative (Household survey, n = 660) and qualitative data collection tools (Focus group discussion, n = 12; In-depth interviews, n = 27) in central Nepal encompassing three altitudinal regions: Lowland (<1000 m amsl; Terai region), Midland (1000–1500 m amsl; hilly region) and Highland (1500–2100 m amsl; mountainous region). We analyzed 37 years (1981–2017) of climatic data from respective districts (Lowland: Chitwan, Dhading; Midland: Kathmandu, Lalitpur; Highland: Nuwakot, Rasuwa). People's perception was compared with climate extreme indices measured along these regions and evaluated if they accurately recognized the impacts on the environment and human health. Our findings show significant climate changes, including rising summer temperature, region-specific winter temperatures and extended monsoon seasons in Nepal. Participants in our study accurately perceived these trends but misperceived heavy precipitation patterns. Reported impacts are rise in crop diseases, human diseases, vector expansion and climate induced disasters like floods, landslides, and water resource depletion, with perception accuracy varying by region. These insights highlight the importance of understanding regional and cross-regional perceptions in relation to climate data in order to develop tailored climate adaptation strategies. Policymakers can use this information to establish region-specific educational and communication initiatives, addressing communities' distinctive vulnerabilities and needs across diverse landscapes. Such approaches can enhance equitable and effective climate resilience in subtropical to alpine regions.

**Funding:** The work was funded by the Federal Ministry of Education and Research of Germany (BMBF) under the project AECO (Number 01KI1717) as part of the National Research Network on Zoonotic Infectious Diseases of Germany. The funders had no role in study design, data collection and analysis, decision to publish, or preparation of the manuscript.

**Competing interests:** The authors have declared that no competing interests exist.

## Introduction

Climate change is undoubtedly the most significant global health threat of the 21st century, exerting both direct and indirect effects on human health, particularly for socially, economically, and culturally vulnerable populations [1–5]. Mountainous regions, such as the Andes, Alps, and the Hindu Kush Himalayan (HKH) region, are especially susceptible to the impact of rising temperatures [6–9]. The consequences of increasing temperature and altered precipitation patterns in these global warming hotspots are manifold, affecting both the environment and public health. Glacial melting, water scarcity, and reduced crop production are prevalent issues in these regions, profoundly impacting the well-being of the local population [7, 9–12]. The HKH region, in particular, is already experiencing notable climate change impacts affecting people's physical and mental health [7, 13, 14]. Thus, unpredictable weather linked to climate change will accelerate the spread of infectious diseases and increase the occurrence of natural hazards and disasters in the HKH region [15]. The urgency of addressing these climate-related health challenges cannot be overstated. This study focuses on Nepal, a developing country located in the HKH region, which has been identified as highly vulnerable to climate change and its impacts [7, 13, 14, 16–20].

Understanding people's perceptions of climate change is crucial when devising policy measures. These perceptions significantly influence their level of concern and subsequently impact their motivation to take action [21, 22]. In order to foster transformative behavioral changes within communities and implement effective and acceptable climate change policies, it is essential to have a comprehensive understanding of people's perceptions [18]. Moreover, studying local concerns and responses within a global warming hotspot can further facilitate global cooperation in mitigating the impacts of climate change [23]. In Nepal, several studies have investigated people's perceptions of climate change, but they all have limitations. Some studies focused solely on perceptions without considering other factors [24–26], while others compared perceptions with a limited set of climate indicators [27–31]. None of these studies incorporated a comprehensive set of climate indicators along with people's perceptions. Instead, studies that included climatic data focused solely on annual mean, minimum, and maximum temperatures and precipitation amounts (detail summary of published literature in S1 Table) [17, 32–39]. However, combining social and extensive climatic data is crucial to evaluate the accuracy of people's perceptions of climatic changes. In this regard, Shrestha et al. [40] were the first to combine perception and climatic data, demonstrating that people in Nepal primarily perceive temperature changes rather than precipitation changes. It is important to note that the relationship between people's perceptions and the reality of climate change is influenced by the specific context. Nepal is divided into different altitudinal regions from South to North, leading to elevation-dependent warming [17]. Thus, those shortcomings from the previous studies motivated us to conduct the current study by coupling objective climate indicators with subjective people's perception to fill the gaps. Therefore, we hypothesize that people's perceptions of climate change and its direct and indirect impacts may vary among these altitudinal regions.

To test the hypothesis regarding variations in people's perceptions across three different altitudinal regions, we employed a cross-sectional mixed-method research design. Our approach involved several steps: 1) we analyzed the annual and seasonal climate trends, including climate extreme indices (detailed in Table 1), and examined the timing of monsoon onset and withdrawal over the past 37 years.

This analysis covered a trend period of 37 years (1981–2017), and we compared the last 30 years (1981–2010) with the most recent 7 years (2011–2017). These investigations were conducted in the Lowland region (<1000 m above mean sea level (amsl)), Midland region (1000–

**Table 1. Definition of temperature and precipitation extreme climate indices [41].**

| Type of indicator | Indices | Indicator name | Definition |
|---|---|---|---|
| **Temperature indicators** | **Percentile indicators** | | |
| | TX90P | Warm days | Percentage of days when TX>90th percentile |
| | TN90P | Warm nights | Percentage of days when TN>90th percentile |
| | TX10P | cool days | Percentage of days when TX<10th percentile |
| | TN10P | cool nights | Percentage of days when TN<10th percentile |
| | **Threshold indicators** | | |
| | SU | Summer days | Annual count when TX(daily maximum)>25˚C |
| | TR | Tropical nights | Annual count when TN(daily minimum)>20˚C |
| | FD | Frost days | Annual count when TN(daily minimum) |
| | **Absolute Indicators** | | |
| | TNx | Warmest nights | Monthly maximum value of daily minimum temp |
| | TXx | Warmest day | Monthly maximum value of daily maximum temp |
| | TNn | Coldest nights | Monthly minimum value of daily minimum temp |
| | TXn | Coldest day | Monthly minimum value of daily maximum temp |
| | DTR | Diurnal temperature range | Daily maximum temperature–daily minimum temperature |
| | **Duration indicators** | | |
| | WSDI | Warm spell duration indicator | Annual count of days with at least 6 consecutive days when TX>90th percentile |
| | CSDI | Cold spell duration indicator | Annual count of days with at least six consecutive days when TN |
| **Precipitation indicator** | **Percentile indicators** | | |
| | R95pTOT | Precipitation on very wet days | Annual total PRCP when RR>95th percentile |
| | **Threshold indicators** | | |
| | R10MM | Number of heavy precipitation days | Annual count of days when PRCP> = 10 mm |
| | R20MM | Number of very heavy precipitation days | Annual count of days when PRCP> = 20 mm |
| | **Absolute indicators** | | |
| | Rx1Day | Monthly max. 1 day precipitation amount | Monthly maximum 1-day precipitation |
| | Rx5Day | Monthly max. 5 day precipitation amount | Monthly maximum 5-day precipitation |
| | **Duration indicators** | | |
| | CWD | Consecutive wet days | Maximum number of consecutive days with RR> = 1 mm |
| | CDD | Consecutive dry days | Maximum number of consecutive days with RR<1m |
| | **Other indicators** | | |
| | PRCPTOT | Annual total wet-day precipitation | Annual total PRCP in wet days (RR>01 mm) |
| | SDII | Simple daily intensity index | Annual total precipitation divided by the number of wet days (defined as Prcp>1 mm) in the year |

1500 m amsl), and Highland region (1500–2100 m amsl) of Central Nepal. 2) We evaluated whether people's perceptions (P) aligned with the actual climate trends. This evaluation utilized both quantitative data from household surveys (HHS) and qualitative data obtained from focus group discussions (FGD) and in-depth interviews (IDI). 3) We assessed whether this association between perceptions and climate trends varied across different altitudes. 4) Additionally, we examined whether people accurately perceived the direct and indirect impacts of climate change, particularly on human health and other sectors, based on their place of residence. Overall, our research design aimed to investigate perceptions across altitudinal regions, compare them with climate trends, and determine the accuracy of perceptions regarding the impacts of climate change on various sectors.

## Materials and methods

### Ethical approval

The Ethical Review Board (ERB) of the Nepal Health Research Council (NHRC), Government of Nepal approved the protocol of this study (registration no. 381/2017). The objectives of the study were explained to the local community people, community leaders and health professionals before the start of the data collection (household surveys, focus group discussions, in-depth interviews). They were informed that participation in the study was voluntary and that they could leave at any time during the interview or withdraw their consent to participate at any point. We obtained written informed consent from all participants.

### Inclusivity in global research

Additional information regarding the ethical, cultural, and scientific considerations specific to inclusivity in global research is included in the Supporting Information (S1 File).

### Study design and setting

In September and October 2018, a cross-sectional mixed-method study (quantitative and qualitative) was carried out in the Lowland, Midland and Highland regions of Central Nepal. Secondary climatic data was collected from the Department of Hydrology and Meteorology (DHM), Nepal. The quantitative data was collected by conducting household surveys (HHS) and the qualitative data by conducting focus group discussions (FGDs) and in-depth interviews (IDIs). This concurrent mixed method design was adopted to triangulate the findings [42]; where, the qualitative data helps to make statistical relations more understandable and intense by using the citations or descriptive language and, thus, this enables a better understanding of the quantitative data [43, 44].

### Study area

Central Nepal (Bagmati province) was selected as study area (Fig 1). The Bagmati province is the largest province of Nepal in regard to the number of inhabitants (about 21% of the total population; [45]. Six administrative districts (Lowland: Chitwan, Dhading; Midland: Kathmandu, Lalitpur; Highland: Nuwakot and Rasuwa) of Central Nepal were selected purposively as they represent broad vertical cross-sections, extending along an altitudinal range from 100 m to 2,100 m above mean sea level (amsl). The study areas (districts) were categorized based on altitudinal variations: Lowland (<1000 m amsl; Terai region), Midland (1000–1500 m amsl; hilly region) and Highland (1500–2100 m amsl; mountainous region) to see whether there were differences between climatic perceptions and real climatic trends at different altitudinal regions. The study sites are connected via road network from South to North. The Lowland and Midland are predominately urban areas with tropical to subtropical climates, compared to the rural Highland, which experiences a temperate to alpine climate [46–48]. Lowland and Midland districts have relatively better concentration of resources, population, physical infrastructure as well as economic and industrial activities compared to Highland districts [49].

### Climate data

To compare people's perception on climate change with the instrumental data, we analyzed 37 years of climate data (temperature, precipitation) from nearby stations of the respective districts within three regions (S2 Table; two weather stations per region: Lowland, Midland, Highland). Firstly, 30 years (1981–2010) of weather data was used as the base period and a

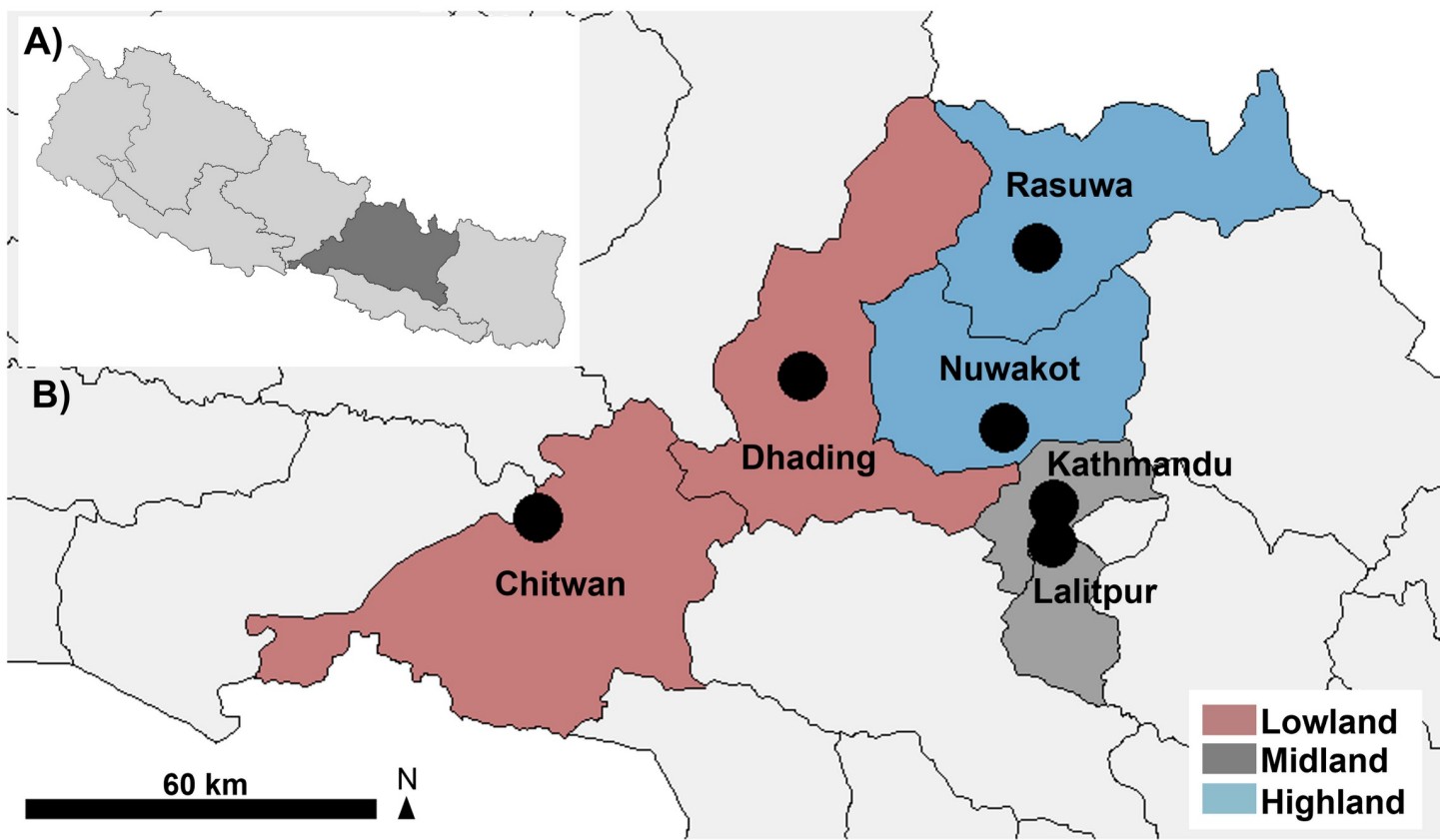

**Fig 1. Map of the study area, Nepal.** (A) Bagmati Province (in dark grey) where the altitudinal gradient is located. (B) Study districts from Lowland (Chitwan, Dhading) and Midland (Kathmandu, Lalitpur) to Highland (Nuwakot, Rasuwa) between 100 to 2100 m amsl. The study sites are indicated with black dots. The map was created using RStudio 2024.04.02 with R packages ggplot2, sf, dplyr and ggspatial.

period of seven years (2011–2017) of weather data was used as the recent time frame to match with the people's perceptions and, secondly, 37 years (1981–2017) of trend analysis was also performed. Climate data was procured from the Department of Hydrology and meteorology (DHM). The approximate distance between the weather stations and social sampling sites are provided (S2 Table).

Annual and seasonal (pre-monsoon, monsoon, post-monsoon, winter) climate extreme indices were calculated using the Climdex software based on the CCl/CLIVAR/JCOMM Expert Team on Climate Change Detection and Indices (ETCCDI; http://etccdi.pacificclimate.org/, Table 1). Suspicious or erroneous data were replaced with missing values before the calculation of the extreme indices (S3 Table), however, for a few of the years, the climate extreme indicator values could not be calculated (S3 Table). Seasonal and annual climate extreme indices were graphically illustrated showing the last 30 years vs. the last 7 years using violin plots (S1–S3 Figs). The indices SU, TR, FD, WSDI and CSDI were only illustrated/calculated annually (details in Table 1). A trend analysis over the last 37 years was conducted to describe the overall annual and seasonal trends of the climatic changes in three altitudinal regions and across those regions in Central Nepal, respectively. The Mann-Kendall test and Sen's slope methods were used to calculate the magnitude and significance of the overall trend in the climate time series data from 1981 to 2017 using an MS-Excel tool called MAKESEN'S (version 1.0) developed by the Finnish Meteorological Institute (FMI) in 2002 [50]. In addition, the

distributions of the annual and seasonal climate extreme indices calculated for the last 30 years (1981–2010) vs. the last 7 years (2011–2017) were compared by means of nonparametric Mann-Whitney tests. Mann-Whitney U-statistics were used to assess the observed versus the perceived climatic changes in different altitudinal regions and across the regions of Central Nepal. The software Prism v.9 (GraphPad, San Diego, CA, USA) was used for all graphical illustrations and statistical analyses.

## Social survey

**Quantitative data collection methods and sampling strategy.** Quantitative data was collected by conducting 660 households surveys from six districts (Lowland: Chitwan and Dhading; Midland: Kathmandu and Lalitpur; Highland: Nuwakot and Rasuwa). In each district, clusters including at least 100 private households were randomly selected for completing a household questionnaire survey (S2 File). Since we did not have a sampling frame, we assume our prevalence as 50% to maximize our sample size. With a 95% confidence level and 5% allowable error, the sample size in each district was 96. i.e., n = z^2*P*Q/d^2 = (1.96^2*0.5^2)/ (0.1^2) = 96. After adding 10% for non-responses, our sample size in each district became 106, which was rounded to 110 for convenience and, thus, in total, 660 households (220 in each region: Lowland, Midland and Highland) were targeted for the household survey.

The households for HHS were chosen in selected clusters along main roads with a 50 m radius of transect employing a simple random sampling. All eligible individuals (aged 18 or above and who had not moved away or died) were listed for each selected household and one participant from that list was selected randomly to take part in the survey using the WHO-Kish method [51]. A questionnaire for climate change perception previously used [43] was adapted for this study. We collected primary data on (1) demographic information (age, education, occupation, marital status, income, ethnicity and the type of residence of the participant), (2) perceptions on the environmental and climate change, and (3) perceptions on the impacts of climate change on human health and other sectors.

**Quantitative data analysis.** The collected quantitative data was verified and entered in the Epi Data 3.1 Software (EpiData Association, Denmark) and analyzed using the statistical package for the Social Sciences software (IBM SPSS Statistics for Windows, Version 24). Chi-square tests were used to compare the socio-demographic characteristics and climate change perceptions between the Lowland, Midland and Highland. The Fisher's exact test was used when appropriate, i.e., when more than 20% of the cells had an expected frequencies of <5 [52].

**Qualitative data collection methods and sampling strategy.** Twelve focus group discussions (FGD) with community people of the respective districts and 27 in-depth interviews (IDIs) with local political leaders, community leaders, female community health volunteers (FCHVs), teachers and public health professionals were carried out using a purposive sampling method [53]. Based on the principal of saturation, the number of IDIs and FGDs were determined [54, 55]. The FGDs and IDIs were conducted in the Nepali language by following the semi-structured guidelines for interview and FGD guide (S3 File); information from these discussions and interviews was recorded. In some cases (n = 2), interviews were not recorded due to unexpected technical problems and only notes were prepared.

**Qualitative data analysis.** Qualitative data analysis was performed using MAXQDA software. First, FGDs and IDIs were transcribed in Nepali language and later translated into English. To avoid bias in the translation and to validate the information, the translation was double-checked by two study team members and then the English version of each transcript

was uploaded into MAXQDA. Initially, themes and sub-themes were defined based on findings from the literature and were then used to create a 'code list' on climate change and its impacts. Emerging themes from the transcripts were also incorporated. Using the code list, the data were coded and recoded in the MAXQDA software, following the approach used in previous study [43]. The themes and subthemes were identified using a simultaneous deductive and inductive approach [43, 44, 56]. The English transcripts were then coded with defined categories accordingly. Finally, all coded material per category was summarized and findings were derived.

## Comparison of people's perception and climatic trends

We compared the perceptions of the people with the climate extremes by means of heat maps (software: Prism v.9, Version 9, GraphPad Software Inc., San Diego-CA, USA). The heat maps (Figs 3 and 4) show the most common answer of the people per perception (given as a fraction of the total answers) and the trend of the climate extreme indices (Sen's slope and significance of the Mann-Whitney U-statistics after FDR correction). We merged the sub-categories for perceptions P1-P6 (S4 Fig) into main categories: P1) "very low" and "low" were summarized as "decrease", P2) "normal high" and "very high" as "increase", and P4) "less shifting" and "much shifting" were merged to "shifting earlier", and, "less prolonging" and "much prolonging" were merged and categorized as "prolonging". Detailed results on P3 were present in P4 and P5 and therefore P3 was not included in the heat map (Fig 4).

The false discovery rate (FDR) was conducted per region and per perception using the 'Two-stage step-up method of Benjamini, Krieger and Yekutieli' with a desired FDR of 5% (for details see Figs 3 and 4). We also illustrated the monthly precipitation trends in the last 30 years vs. the last 7 years. In addition, to compare the people's perceptions of the shift or prolonging of the monsoon season with timely data, the dates of the monsoon onset, the withdrawal of the monsoon and the monsoon season length were downloaded from the Department of Hydrology and Meteorology, Kathmandu Nepal (http://dhm.gov.np/download/) for the whole of Nepal (region-specific values were not present). The onset of the monsoon in Nepal was determined by the Department of Hydrology and Meteorology according to the following factors: 1) the wind direction over South-Eastern Nepal at the surface and upper levels should be south-eastern at the surface with a western direction jet wind towards the Tibetan Plateau, 2) past three days of continuous rainfall should be present, 3) the air pressure at the surface and upper levels should be a low pressure system, 4) the progression of monsoon onset from the South to North from India is evaluated, and 5) outgoing longwave radiation is investigated and should be reduced over Eastern Nepal. The monsoon onset and withdrawal dates as well as the monsoon season length were analysed over time (37 years) using the Mann-Kendall test and the Sen's slope methods. In addition, the last 30 years vs. 7 years were compared with each other using a Mann-Whitney test.

## Results

The climate in Central Nepal has undergone significant changes in recent decades, including increasing summer temperatures, region-specific winter temperature increases, and an extended monsoon season. People have generally perceived these temperature trends accurately. However, there has been a misperception regarding a decrease in heavy precipitation patterns across the regions. People have also reported both direct and indirect impacts of climate change. Across different altitudinal regions, they have particularly noted an increase in crop diseases and the appearance of vectors in new areas. Furthermore, region-specific impacts include floods, landslides, transmission of vector-borne diseases, and the depletion of water

resources. The participants from all study areas correctly perceived the growing threat of vector expansion.

## Comparison of perception on direct climate change impacts and climatic trends

People's perceptions of meteorological and climatic changes and their impact on human health generally aligned with the climate trends observed in the study areas (P = perception; P1, P2 for all areas; P2, P5 for Lowland and Midland; P4 for Midland and Highland), except for heavy rainfall in recent years (P6; Fig 2). In summary, people accurately identified the overall increasing trend in summer temperatures across all altitudinal regions (P1) and the region-specific temperature trends in winter (P2; Table 2, Figs 2 and 3).

In the Lowland, HHS participants' perception on a trend in winter cold was discordant (feels same; 31.5%, decrease; 31.5%, increase; 30.6%), while the IDI and FGD participants

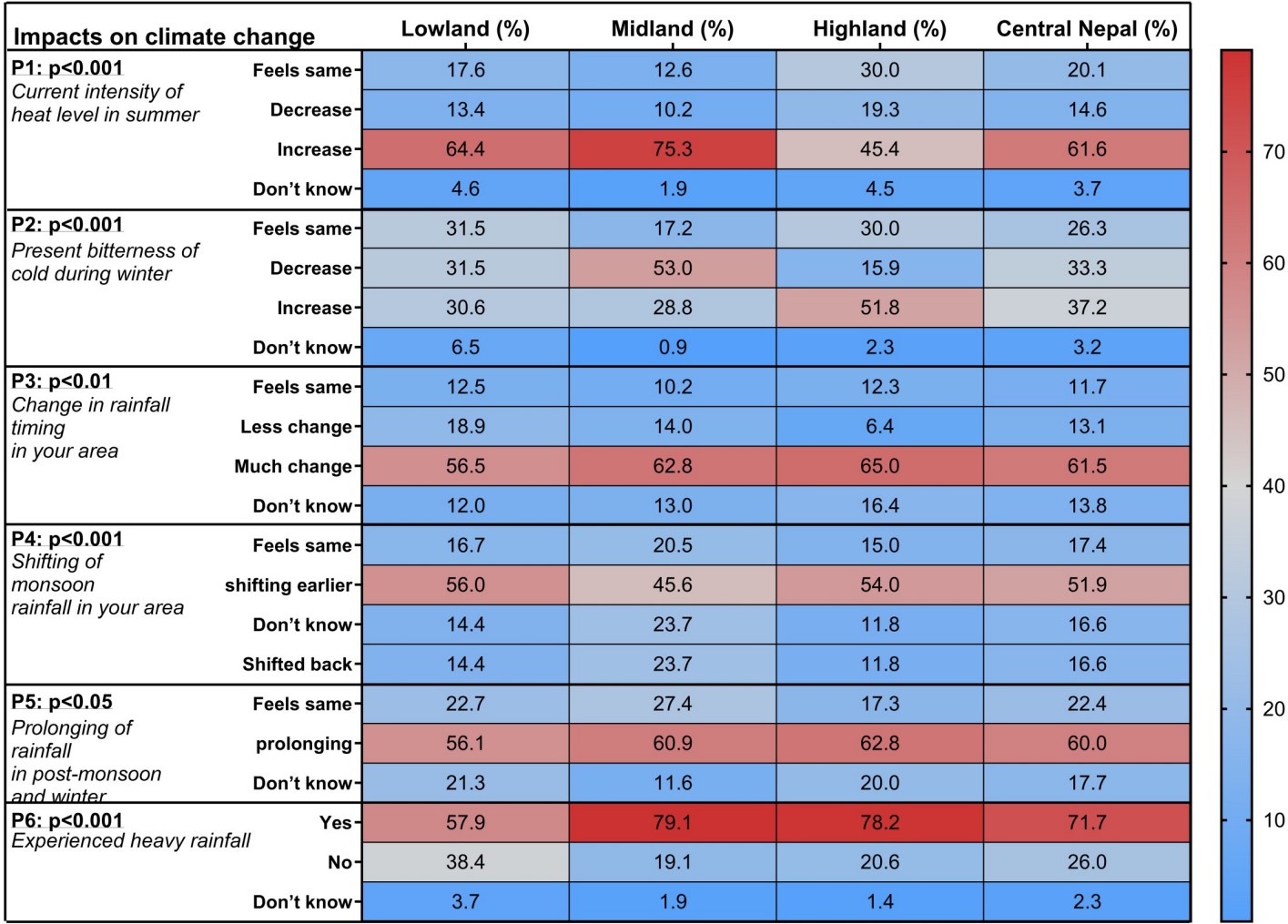

| Impacts on climate change | | Lowland (%) | Midland (%) | Highland (%) | Central Nepal (%) |
|---|---|---|---|---|---|
| **P1: p<0.001** *Current intensity of heat level in summer* | Feels same | 17.6 | 12.6 | 30.0 | 20.1 |
| | Decrease | 13.4 | 10.2 | 19.3 | 14.6 |
| | Increase | 64.4 | 75.3 | 45.4 | 61.6 |
| | Don't know | 4.6 | 1.9 | 4.5 | 3.7 |
| **P2: p<0.001** *Present bitterness of cold during winter* | Feels same | 31.5 | 17.2 | 30.0 | 26.3 |
| | Decrease | 31.5 | 53.0 | 15.9 | 33.3 |
| | Increase | 30.6 | 28.8 | 51.8 | 37.2 |
| | Don't know | 6.5 | 0.9 | 2.3 | 3.2 |
| **P3: p<0.01** *Change in rainfall timing in your area* | Feels same | 12.5 | 10.2 | 12.3 | 11.7 |
| | Less change | 18.9 | 14.0 | 6.4 | 13.1 |
| | Much change | 56.5 | 62.8 | 65.0 | 61.5 |
| | Don't know | 12.0 | 13.0 | 16.4 | 13.8 |
| **P4: p<0.001** *Shifting of monsoon rainfall in your area* | Feels same | 16.7 | 20.5 | 15.0 | 17.4 |
| | shifting earlier | 56.0 | 45.6 | 54.0 | 51.9 |
| | Don't know | 14.4 | 23.7 | 11.8 | 16.6 |
| | Shifted back | 14.4 | 23.7 | 11.8 | 16.6 |
| **P5: p<0.05** *Prolonging of rainfall in post-monsoon and winter* | Feels same | 22.7 | 27.4 | 17.3 | 22.4 |
| | prolonging | 56.1 | 60.9 | 62.8 | 60.0 |
| | Don't know | 21.3 | 11.6 | 20.0 | 17.7 |
| **P6: p<0.001** *Experienced heavy rainfall* | Yes | 57.9 | 79.1 | 78.2 | 71.7 |
| | No | 38.4 | 19.1 | 20.6 | 26.0 |
| | Don't know | 3.7 | 1.9 | 1.4 | 2.3 |

**Fig 2. People's perceptions on climate change and variability.** Heat map of the perceptions P1 to P6 (% participants, p-value from Chi-square test) in accordance with the altitudinal residence of the participants in Lowland, Midland and Highland or their general residence in Central Nepal. Detailed responses of participants are given in S8 Fig (details S4 File).

**Table 2. Comparison of the trend of people's perceptions and the climatic trends (social data: Quantitative data: Household survey (n = 660); qualitative data: Focus group discussions (n = 12), in-depth interviews (n = 27); climate data: Climate extreme indices, Table 1).** In addition, the temporal trend is given for the perceptions related to the monsoon (S9 Fig). Depending on the perception, the presence of a trend (yes, no, unclear/mixed) or a trend direction (increase, decrease, unclear) is given. Significance of each data type is presented in detail in Figs 2–5. Qualitative data trend of people's perception was evaluated by the authors. The trend in quantitative data is always represented by the most frequently occurring value, if it is not clearly different from the others it was given as unclear (Figs 1 and 5). Climatic trend was given by summarizing observed (significant) trends (Figs 3, 4 and S9 Fig). Brackets indicate only a small trend (climate = one or a few significant indices, temporal shift = no significant trend). NA = data not available. Green color = social and climate data is matching. Grey color = one data type is missing, so a comparison is not possible (details S4 File).

| People's perception on climate change | | | | |
|---|---|---|---|---|
| **Perception** | **Data Type** | **Lowland** | **Midland** | **Highland** |
| **P1**: Intensity of heat level in summer | Climate | increase | increase | increase |
| | Quantitative social data | increase | increase | increase |
| | Qualitative social data | increase | increase | increase |
| **P2**: Bitterness of cold during winter | Climate | (decrease) | decrease | decrease |
| | Quantitative social data | decrease | decrease | increase |
| | Qualitative social data | increase | decrease | increase |
| **P3**: Change in rainfall timing | Climate (annual) | yes | yes | yes |
| | Temporal shift | yes | yes | yes |
| | Quantitative social data | yes | yes | yes |
| | Qualitative social data | yes | yes | yes |
| **P4**: Monsoon shifting earlier | Climate (pre-monsoon) | (increase) | (increase) | (increase) |
| | Temporal shift | (yes) | (yes) | (yes) |
| | Quantitative social data | yes | yes | yes |
| | Qualitative social data | no | mixed | yes |
| **P5**: Prolonging of monsoon to post-monsoon and winter | Climate (post-monsoon) | (increase) | (increase) | (decrease) |
| | Temporal shift | yes | yes | yes |
| | Quantitative social data | yes | yes | yes |
| | Qualitative social data | yes | yes | no |
| **P6**: Experienced heavy rainfall in later years | Climate | (decrease) | (decrease) | (decrease) |
| | Quantitative social data | yes | yes | yes |
| | Qualitative social data | yes | yes | yes |
| **People's perception on environmental change** | | | | |
| **Perception** | **Data Type** | **Lowland** | **Midland** | **Highland** |
| **P7**: Experienced drying of water resources | Quantitative social data | no | yes | yes |
| | Qualitative social data | yes | yes | yes |
| **P8**: Experienced increase in frequency of droughts | Quantitative social data | (yes) | (yes) | (yes) |
| | Qualitative social data | yes | yes | no |
| **P9**: Experienced of mosquitoes in new areas | Quantitative social data | yes | yes | yes |
| | Qualitative social data | yes | yes | yes |
| **P10**: Experienced the transmission of vector-borne diseases in new areas | Quantitative social data | yes | yes | no |
| | Qualitative social data | yes | yes | no |
| **P11**: Experienced of new human diseases | Quantitative social data | no | no | no |
| | Qualitative social data | yes | yes | yes |
| **P12**: Experienced new crop diseases | Quantitative social data | yes | yes | yes |
| | Qualitative social data | yes | yes | yes |
| **P13**: Experienced new domestic animal diseases | Quantitative social data | no | no | no |
| | Qualitative social data | NA | NA | NA |
| **P14**: Experienced less snow | Quantitative social data | yes | yes | yes |
| | Qualitative social data | NA | NA | NA |
| **P15**: Experienced increase in floods and landslides | Quantitative social data | no | no | yes |
| | Qualitative social data | yes | yes | yes |

| P1: Current intensity of heat level in summer in comparison to last 5 to 10 years | | | | |
|---|---|---|---|---|
| | | Lowland (%) | Midland (%) | Highland (%) | Central Nepal (%) |
| summer/monsoon season | Increase | 0.644 | 0.753 | 0.454 | 0.616 |
| | TR⁺ | 0.690* | **1.000***** | ✕ | 0.400 |
| | SU⁺ | -0.170 | 1.160*** | 0.640** | -0.350 |
| | DTR | 0.004 | 0.021* | 0.022 | 0.009 |
| | TNx | 0.032*** | **0.012*** | 0.016 | -0.002 |
| | TXx | 0.017 | **0.043***** | 0.030** | 0.026*** |
| | TX90p | 0.199* | **0.455***** | **0.378**** | **0.339***** |
| | TN90p | 0.224** | **0.347***** | 0.264* | 0.257*** |
| | WSDI⁺ | -0.040 | 0.110 | 0.000 | 0.080 |

| P2: Present bitterness of cold during winter in comparison to last 5 to 10 years | | | | |
|---|---|---|---|---|
| winter | Decrease | 0.315 | 0.530 | 0.159 | 0.333 |
| | Increase | 0.306 | 0.288 | 0.518 | 0.372 |
| | Feel same | 0.315 | 0.172 | 0.300 | 0.263 |
| | FD⁺ | -0.030 | **-0.300***** | 0.000 | **-0.150** |
| | DTR | -0.016 | **-0.006** | -0.057* | -0.032* |
| | TXn | -0.020 | 0.080*** | 0.050*** | 0.000 |
| | TNn | 0.000 | **0.075***** | 0.057** | 0.030* |
| | TX10p | 0.013* | -0.355*** | -0.144** | -0.170*** |
| | TN10p | -0.078* | **-0.508***** | -0.387* | -0.317*** |
| | CSDI⁺ | 0.000 | **-0.290***** | -0.060*** | -0.160* |

⁺annual, all other values seasonal

**Fig 3. The match between the perceptions P1-P2 and the seasonal precipitation/climate parameters.** Heat map of the perceptions P1 and P2 (given as fractions of the total answers) and the trend (Sen's slope) over the last 37 years (1981–2017) of the summer/monsoon season and winter-related climate extreme indices (Table 1) in accordance with the altitudinal residence of participants in Lowland, Midland and Highland or their general residence in Central Nepal. Significant climate indicators from the trend analysis (Mann-Kandall test) over the last 37 years (1981–2017) are marked with a asterisk (* p < 0.05, ** p < 0.01, *** p < 0.0001). Significant climate indicators (Mann-Whitney U-test) in the 30 years (1981–2010) vs. 7 years (2011–2017) comparison, after FDR correction, are marked in bold. In the Highlands, no tropical nights (TR) were recorded (details S4 File).

reported an increase in winter cold (Table 2, Fig 3). In line, a clear significant temperature trend in the Lowlands was not present (cool days (TX10P) increase, cool nights (TN10P) decrease). The majority of participants in the Midland correctly perceived a decreasing trend in winter cold, which aligns with observed climatic trends. In the Highland, the perception of increasing winter cold was partly discordant with the observed climatic trend of a decrease in winter cold over the past 37 years, although some data in the Highland were missing in the past 30 years (Fig 3, S3 Table). Additionally, the number of frost days did not significantly decrease in the Highland, unlike the Midland where all indices showed significant differences in the past 7 to 30 years, and temperature extremes such as the number of cold days/nights were increasing. These factors potentially explain why people perceived an increase in winter cold in the Highland. The temperature trend in winter in the Lowland, as well as partly in the

Highland, was not as systematic and highly significant as in the Midland, which may contribute to the varying perceptions among people.

People correctly perceived a change in the monsoon rainfall pattern (P3, Fig 4, Table 2, and S9 Fig). Although there was no clear systematic shift in precipitation-related climate extreme indices (only small trends existed), the majority of participants from all altitudinal regions consistently perceived temporal changes, such as an earlier start and prolonged monsoon season, and increased heavy rainfall (P4, P5, P6; Fig 4).

Perceptions P3 and P5 align with the observed withdrawal of the monsoon and an increase in the number of monsoon days (S9 Fig). However, when comparing climate data with quantitative and qualitative data on the timing of monsoon rainfall (P4, P5), discrepancies were found, indicating that people struggle to accurately identify temporal precipitation trends, such as earlier/later onset and withdrawal (Table 2). Perception P6 showed a complete discrepancy between people's perception and climate data (Table 2, Fig 4). People perceived increasing rainfall in all regions of Central Nepal, despite the fact that the number of heavy precipitation days was decreasing (Table 2, Fig 4).

## Comparison of perception on indirect climate change impacts and climatic trends

People in Central Nepal also perceived various indirect impacts of climate change (P7-P15; Table 2, Fig 5). Direct impacts, such as the drying-up of water resources (P7), were particularly perceived in the Midland and Highland. Although people could not directly perceive the decreasing trend in precipitation, they indirectly noticed the drying-up of water resources, which indicated this negative trend (P7; Fig 5).

Additionally, an increase in floods and landslides was mostly reported in the Highland (HHS), while participants in the Lowland and Midland (IDI and FGD) partly reported this increase. Correspondingly, heavy precipitation (Rx5Day) and the number of wet days (CWD) increased during the monsoon season in the Highland, potentially contributing to such events. In the Midland, heavy precipitation events (Rx5Day) and overall annual heavy precipitation (R95pTOT) also increased during the monsoon and post-monsoon seasons. Furthermore, people from each region perceived a decrease in snowfall days and snow cover (P14), as well as an increase in the duration of droughts and drought-like conditions (P8) (Table 2, Fig 5). This increase in droughts can be linked to a decreasing trend in annual rainfall patterns (R10mm, R20mm, Rx5day, CDD; S9 Fig) observed in all regions.

Regarding indirect impacts, people in the Lowland and Midland perceived the presence of mosquitoes (P9) and the transmission of vector-borne diseases in new areas (P10), while in the Highland, people only perceived the spread of mosquitoes to new areas (Table 2, Fig 5). In addition, people (IDIs) reported the new occurrence of poisonous snakes (Cobra, Krait) in the Highland areas in recent years (details in S4 File). People from all regions perceived an increase in crop diseases (P12), but no new human diseases (P11) were reported (Table 2, Fig 5). However, FGD and IDI participants from all regions reported a rise in non-communicable diseases, including diabetes and hypertension. Additionally, they noted that seasonal flu has become more severe and spreads more rapidly, with common colds now requiring longer recovery times despite previously being manageable with simple remedies (S4 File). In general, participants (HHS, FGD, and IDI) were able to distinguish between vector-borne diseases and other human diseases, although there were differing perceptions regarding the occurrence of human diseases (HHS: no increase; IDI, FGD: increase; Fig 5)382. Similarly, the IDI participants from all regions (Lowland, Midland, Highland) have also reported that unusual rainfall patterns and climate-induced disasters along with crop pests had adversely affected the production of food grains, vegetables and fruits.

| P4: Shifting of monsoon rainfall in your area earlier than past 5 to 10 years | | | | |
|---|---|---|---|---|
| | | Lowland (%) | Midland (%) | Highland (%) | Central Nepal (%) |

**pre-monsoon**

| | Lowland (%) | Midland (%) | Highland (%) | Central Nepal (%) |
|---|---|---|---|---|
| shifting earlier | 0.560 | 0.456 | 0.540 | 0.519 |
| CDD | -0.125 | -0.115 | -0.071 | -0.119 |
| CWD | 0.000 | -0.025 | 0.075** | 0.016 |
| Rx1Day | -0.002 | 0.200 | -0.050 | 0.058 |
| Rx5Day | 0.036* | **0.004** | 0.013 | 0.016 |
| R10mm | 0.000 | 0.000 | -0.093 | -0.026 |
| R20mm | 0.000 | 0.033 | -0.023 | 0.014 |

**monsoon +**

| | Lowland (%) | Midland (%) | Highland (%) | Central Nepal (%) |
|---|---|---|---|---|
| CDD | -0.047 | -0.045 | -0.075 | -0.050 |
| CWD | -0.115 | -0.095* | 0.833*** | 0.241* |
| Rx1Day | -0.077 | 0.150 | -0.066 | 0.039 |
| Rx5Day | -0.167 | 1.164*** | 0.638** | -0.211 |
| R10mm | 0.016 | -0.067 | 0.250 | 0.099 |
| R20mm | 0.048 | 0.000 | 0.068 | 0.059 |

| P5: Prolonging of rainfall in post-monsoon and winter than past 5 to 10 years | | | | |
|---|---|---|---|---|

**post-monsoon**

| | Lowland (%) | Midland (%) | Highland (%) | Central Nepal (%) |
|---|---|---|---|---|
| Prolonging | 0.561 | 0.609 | 0.628 | 0.600 |
| CDD | 0.167 | 0.035 | 0.282 | 0.183 |
| CWD | 0.000 | 0.000 | 0.000 | 0.000 |
| Rx1Day | -0.200 | 0.011 | -0.239 | -0.152 |
| Rx5Day | 0.686* | 1.000*** | -0.208* | 0.351* |
| R10mm | 0.000 | -0.016 | -0.083* | -0.047* |
| R20mm | 0.000 | 0.000 | 0.000 | -0.015 |

**winter**

| | Lowland (%) | Midland (%) | Highland (%) | Central Nepal (%) |
|---|---|---|---|---|
| CDD | 0.387* | 0.246 | 0.250 | 0.342 |
| CWD | 0.000 | 0.000 | 0.000 | 0.006 |
| Rx1Day | -0.073 | -0.117 | -0.063* | -0.087 |
| Rx5Day | -0.008 | 0.000 | -0.032 | -0.033** |
| R10mm | 0.000 | 0.000 | -0.077* | -0.039 |
| R20mm | 0.000 | 0.000 | 0.000 | 0.000 |

| P6: Experienced heavy rainfall in later years. | | | | |
|---|---|---|---|---|

**annual**

| | Lowland (%) | Midland (%) | Highland (%) | Central Nepal (%) |
|---|---|---|---|---|
| Yes | 0.579 | 0.791 | 0.782 | 0.717 |
| PRCPTOT | -1.33 | -0.96 | 7.53 | -1.33 |
| R10mm | -0.06*** | -0.02* | -0.01* | -0.06 |
| R20mm | -0.01* | -0.14** | -0.07*** | 0.01 |
| R95pTOT | -0.36 | 1.59* | 0.08 | 0.86 |
| SDII | 0.02 | 0.02 | -0.09* | -0.02 |

Scale legend: above scale; 1.5; 1.0; 0.5; 0; -0.5; -1.0; below scale

+not used for FDR correction

**Fig 4. The match between the perceptions P4-P6 and the seasonal precipitation/climate parameters.** Heat map of the perceptions P4 –P6 (given as fractions of the total answers) and the trend (Sen's slope) over the past 37 years (1981–2017) of seasonal and annual precipitation-related climate extreme indices (Table 1) in accordance with the altitudinal residence of the participants in Lowland, Midland and Highland or their general residence in Central Nepal. Significant indicators from the trend analysis (Mann- Kendall test) over the past 37 years (1981–2017) are marked with an asterisk (* $p < 0.05$, ** $p < 0.01$, *** $p < 0.0001$). Significant climate indicators (Mann-Whitney U-test) in the 30 years (1981–2010) vs. 7 years (2011–2017) comparison, after FDR correction, are marked in bold. For P4, the results of the monsoon season are given to complete the dataset, however, the monsoon season was not part of the FDR correction for P4 (details S4 File).

| Impacts on climate change | | Lowland (%) | Midland (%) | Highland (%) | Central Nepal (%) |
|---|---|---|---|---|---|
| **P7: p<0.001** *Experienced drying of water resources in later years* | Yes | 40.7 | 80.0 | 55.9 | 58.8 |
| | No | 50.0 | 14.4 | 40.5 | 35.0 |
| | Don't know | 9.3 | 5.6 | 3.6 | 6.1 |
| **P8: p>0.05** *Experienced increase in frequency of droughts in later years* | Yes | 53.7 | 51.6 | 49.5 | 51.6 |
| | No | 41.2 | 46.0 | 45.0 | 44.1 |
| | Don't know | 5.1 | 2.3 | 5.5 | 4.3 |
| **P9: p<0.001** *Experienced of mosquitoes in new areas not present before* | Yes | 61.9 | 65.6 | 77.3 | 68.3 |
| | No | 20.9 | 27.4 | 19.1 | 22.5 |
| | Don't know | 17.2 | 7.0 | 3.6 | 9.2 |
| **P10: p<0.001** *Experienced the transmission of VBDs in new areas* | Yes | 38.9 | 42.8 | 33.6 | 38.4 |
| | No | 32.4 | 41.9 | 49.5 | 41.3 |
| | Don't know | 28.7 | 15.3 | 16.8 | 20.3 |
| **P11: p<0.001** *Experienced new human diseases in later years compared to past* | Yes | 19.9 | 20.0 | 8.6 | 16.1 |
| | No | 44.0 | 46.0 | 63.2 | 51.2 |
| | Don't know | 36.1 | 34.0 | 28.2 | 32.7 |
| **P12: p<0.01** *Experienced new crop diseases in later years compared to past* | Yes | 48.1 | 45.1 | 42.3 | 45.2 |
| | No | 24.1 | 19.5 | 33.6 | 25.8 |
| | Don't know | 27.8 | 35.3 | 24.1 | 29.0 |
| **P13: p<0.001** *Experienced new domesic animal diseases in later years compared to past years* | Yes | 33.8 | 24.7 | 34.1 | 30.9 |
| | No | 30.6 | 22.2 | 37.3 | 30.1 |
| | Don't know | 35.6 | 53.0 | 28.6 | 39.0 |
| **P14: p>0.05** *Experienced less snow in Himalayas in later years compared to the past* | Yes | 37.5 | 43.7 | 49.5 | 43.6 |
| | No | 28.7 | 28.4 | 26.8 | 28.0 |
| | Don't know | 33.8 | 27.9 | 23.6 | 28.4 |
| **P15: p<0.001** *Experienced increase in floods and land-slides in later years compared to past* | Yes | 38.9 | 36.7 | 53.2 | 43.0 |
| | No | 44.0 | 51.2 | 38.2 | 44.4 |
| | Don't know | 17.1 | 12.1 | 8.6 | 12.6 |

**Fig 5. People's perceptions on the impacts of environmental and climate change.** Heat map of the perceptions P7–15 (percentages of the answers by the participants are given; p-values from the Chi square test) in accordance with the residential area of the participants in Lowland, Midland and Highland or their general residence in Central Nepal.

## Discussion

Mountainous regions, such as the Hindu Kush Himalayan region, are facing rapid climate change and are particularly vulnerable to its direct and indirect impacts [6–9, 57, 58]. In our study, we investigated people's perceptions of climate and climate change in different altitudinal regions of Nepal, which is part of the HKH region. Our findings provide valuable insights into how people perceive climate warming and its impact on human health. We demonstrated that observed climatic trends in Central Nepal are partially perceived accurately, although this perception varies depending on participants' residential areas and the extent of climatic changes in those areas. Understanding the direct and indirect impacts of climate change, both regionally and cross-regionally, can contribute to fostering public engagement and developing effective communication and educational strategies regarding the connections between climate warming and health [18]. Furthermore, our results have global implications, serving as a forecast for major aspects of climate change that will likely be perceptible to people in the future from subtropical to alpine regions.

Our study on people's perceptions of climate change revealed that participants accurately perceived the seasonal temperature trends in different altitudinal regions, particularly during summer and winter. In various areas of the HKH region, increasing temperatures were perceived and sometimes supported by the observed temperature trends (detailed comparison of people's perception and temperature trends in multiple regions of the HKH region are summarized in S1 Table). Consistent with Central Nepal, an overall increase in summer temperatures was commonly perceived in the Lowland and Midland regions of the entire HKH region, which aligns with previous studies conducted in other parts of Eastern, Western and Central Nepal, and in other HKH countries like Bangladesh and India [25, 27–29, 31, 59, 60]. An earlier onset of summer was perceived in general in Central Nepal and specifically by the Highland communities, which is in line with previous studies from Eastern Nepal [31, 61]. Likewise previous research conducted in Nepal [31], we found the Midland and Highland communities perceived shorter winters, while a decrease in winter cold was perceived in the Lowland of Central Nepal which is consistent with previous studies in Western Lowland parts of HKH country, Nepal [27, 28]. In Central Nepal, the trend of decreasing winter cold was most pronounced in the Midland, likely influenced by urbanization and the heat island effect [37, 62–65]. Conversely, people in the Highland of Central Nepal perceived an increase in winter cold, while in other HKH regions, this perception was only reported in the Lowland [66].

Our findings confirmed that in Nepal, changes in precipitation patterns and heavy precipitation trends are not as noticeable as temperature changes [40]. The majority of people in the Highlands perceived an increase in heavy precipitation, which may be related to the rising trend of rainfall in those regions (PRCPTOT: 7.53mm/year, not significant). Since a significant portion of Highland people engage in agriculture (33.5%; S5 Fig), they are likely to notice even small changes in precipitation [67]. Overall, people in the HKH region perceived changes in the monsoon rainfall pattern, particularly in terms of timing, although previous studies have shown that perceptions of precipitation changes vary across different regions in the HKH region [24–29, 31, 40, 59, 60, 66, 68]. In recent years (2005–2016), Nepal has experienced an increase in indirect climate change impacts such as floods and landslides [58, 69]. This increase may have led people in Central Nepal to perceive an increase in heavy rainfall, even though heavy rainfall had actually decreased. From 2000–2009, compared to each other, Lowland districts were highly affected by floods (including Chitwan, also in 2017), while Highland districts were moderately and the Midland districts were less affected [69, 70]. Highland residents in our study may have perceived more floods due to the increasing frequency of outburst floods from glacial lakes [71]. Additionally, landslides have heavily impacted the Nuwakot district, which is part of the Highland region [70, 72]. Indeed, loss of life and property due to extreme rainfall events has already been reported in Nepal [69, 70, 73]. Such negative events have a significant impact on the psychological well-being of individuals [74], potentially explaining why people perceived an increase in heavy rainfall patterns. In general, people in the HKH region perceived an increase in floods across all altitudinal regions, while an increase in landslides was also frequently reported in the Midland and Highland regions, in line with previous studies [24, 29, 31, 40, 57, 59, 68]. In fact, a study in Nepal has also reported increasing frequency of climatic disasters and, mortality due to these climate induced disasters since 3 decades (1992–2021), highlighting the vulnerability of the Mid hills (Midland) and Mountains regions (Highland) to landslides and the Lowland Terai regions to floods [57].

Furthermore, droughts in Central Nepal have become more severe and frequent [75], and this increase has been perceived by people across all regions. Dryness or droughts have also been perceived in other parts of the HKH region, highlighting the significant impact of global warming [24, 29, 31, 40, 59, 68]. The agricultural sector, including agricultural production, is affected by increasing temperatures and droughts caused by climate warming [7], reflecting

our findings. Additionally, the people from all regions of Central Nepal perceived an increase in new crop diseases and the reduction in crop production. These findings are consistent with previous studies in other parts of Western and Central Nepal where people also reported declining of crop yield due to increase in new crop pests along with climate-induced disasters [24, 31, 76]. This might be due to rising temperatures that influence reproduction, spread and severity of numerous plants pathogens, while also causing a shifting of crop pathogens from Lowland to Highland regions [77, 78]. Additionally, people in our study have also observed changes in biodiversity, such as early flowering of plants, which is consistent with previous studies conducted in Nepal [31, 68]. Similarly, people residing in Pakistan have also reported changes in plant distribution (shift to higher altitudes), abundance, and flowering periods due to climate change, aligning with our findings from FGDs in Central Nepal [79].

At the interface of biodiversity and health, health-threatening animals, such as venomous snakes, are expanding their distribution to higher altitudes as perceived by the IDI participants in our study and as demonstrated by previous studies conducted in Nepal [80, 81]. Another health threat to humans is the altitudinal expansion of vector-borne diseases (e.g., malaria and dengue) throughout the entire HKH region [14]. Vector-borne diseases and their vectors are already common in the Low- and Midland regions (Fig 4- P4 and 5; [14, 16, 82, 83]), while climate change-induced vector-borne diseases, such as dengue, are projected to increase and become more common at higher altitudes [14, 84]. Therefore, in the present study people from Central Nepal, especially in Lowland regions, have correctly reported an increase or emergence of vectors [24, 29, 66, 68]. In the Lowland region of Central Nepal, FGD and IDI participants have experienced an increase in communicable (influenza) and non-communicable diseases (diabetes, blood pressure). This observation aligns with research indicating a rise in influenza in Nepal since 2004 [85, 86], as well as predictions that non-communicable diseases will worsen across the HKH region [7].

In general, people's motivation to take action on climate change is influenced by their level of concern, which is influenced by their individual perceptions [21, 22]. Hence, increasing the local, regional and cross-regional adaptive capacity will help to reduce the impacts of climate change on the health and well-being of people residing in the Nepal specifically and the HKH region in particular [7].

## Conclusion and recommendations

This study could offer valuable insights into people's perceptions of climate change in relation to weather data, not only within the HKH region but also in other regions worldwide with similar geographical landscapes. Our study provides also important insights into people´s perceptions on direct and indirect impacts of climate changes on human health, agriculture, biodiversity and overall in environment in Central Nepal. The observed climatic trends in Central Nepal are in part correctly perceived, which strongly depends on the residential area of participants and the respective extent of climate changes. Thus, the level of understanding of the direct and indirect impacts of climate change encompasses some regional and some cross-regional aspects that could be helpful to build widespread public engagement and develop effective communication and educational approaches on the interactions of climate change and health in a region-specific manner. Globally, our findings provide a projection of significant aspects of climate change that will likely be perceptible to people in the future. The perceptions of Nepalese people and those living in the HKH region underscore the urgent need for coping strategies to address declining agricultural productivity, prompting a recommendation for local stakeholders and regional governments to focus on measures such as improved water management practices for the dry season, adjusted sowing and planting

schedules, modified cropping patterns, and environmentally sustainable professional pest control methods which are already being in practiced by farmers in some parts of Nepal [87]. Also, countrywide biodiversity conservation strategies, combined with climate change mitigation and adaptation activities, are necessary, especially to mitigate the increasing heat. It is also advisable to revise plans for coping with the impacts of climate change, especially for victims of natural disasters, according to region-specific perceptions. Similarly, comprehensive information and awareness campaigns addressing the risk of snakebites in highland regions, alongside initiatives to enhance public understanding of vectors and vector-borne diseases, should be implemented across all regions to encourage the adoption of effective preventive measures [88]. Meanwhile, strengthening Nepal's public health infrastructure and establishing early warning systems for climate-sensitive and epidemic-prone infectious diseases in the HKH region are essential to prevent further spread of vectors and vector-borne diseases. [7, 89]. Thus, we recommend that policymakers in Central Nepal and in general in the HKH region focus on designing, communicating, and implementing climate adaptation strategies [90] based on the direct and indirect climate change impacts that were primarily perceived in this study. Those strategies should address both regional and cross-regional climate change impacts.

## Strengths and limitations

The strength of the present study lies in its cross-sectional mixed-method research design, which combines different data sources on climate change and perception, including climatic data and social data, sampled along an altitudinal gradient. This research design provided us with the opportunity to triangulate the findings and gain a deeper understanding of the relationship between climate change and people's perceptions. However, this study must be interpreted with caution regarding certain aspects. The data was collected in densely populated urban and semi-urban areas of each altitudinal region, and therefore, the clustering of households within a 50 m radius around the data collection site may not be representative of the districts and the entire country. Additionally, our study is focused exclusively on Central Nepal (Bagmati province), the most populated region of Nepal. Considering the country's significant geographical and socio-economic diversity, conducting similar studies in other regions of Nepal in the future could provide a more comprehensive understanding of climate variability and its impacts across diverse geographical and socio-economic contexts. However, we have also extensively reviewed the relevant literature to identify similarities and differences between the different altitudinal regions in Central Nepal and other studies conducted in general in the HKH region (S1 Table).

   This is the first study that has identified region-specific as well as cross-region specific impacts of climate change in the HKH region itself. The study underlines that transdisciplinary research is essential and the first step to plan later effective implementation of mitigation and adaptation strategies that would be accepted by the public in the face of climate change impacts. A better understanding of people's perceptions of climate change in a global warming hotspot will likely help policymakers induce transformational behavioral changes not only in the HKH region but also globally.

## Supporting information

**S1 Fig. Annual climate extreme indices related to warm temperatures in Central Nepal.** Results are given for the Lowland, Midland and Highland and also across the altitudinal gradient in Central Nepal during a past 30 year's period (1981–2010) vs. past 7 years (2011–2017). Climatic indices: A) TX90P- warm days, B) TN90P- warm nights, C) SU-summer days, D)

TR-Tropical nights, E) TXx- warmest day F) TNx-warmest night and G) WSDI- warm spell duration indicator. Significant climate indicators (Mann-Whitney U-test) in the 30 years (1981–2010) vs. 7 years (2011–2017) comparison after FDR correction are marked with an asterisk.
(TIF)

**S2 Fig. Annual climate extreme indices related to cold temperatures in Central Nepal.** Results are given for the Lowland, Midland, Highland and also across the altitudinal gradient in Central Nepal during a past 30 year- period (1981–2010) vs. past 7 years (2011–2017). Climatic indices: A) TX10P- cool days, B) TN10P- cool nights, C) FD- frost days, D) DTR- daily temperature range, E) TXn- coldest day F) TNn- coldest night and G) CSDI- cold spell duration indicator. Significant climate indicators (Mann-Whitney U-test) in the 30 years (1981–2010) vs. 7 years (2011–2017) comparison after FDR correction are marked with an asterisk.
(TIF)

**S3 Fig. Annual precipitation-related climate extreme indices in Central Nepal.** Results are given for the Lowland, Midland and Highland and also across the altitudinal gradient in Central Nepal during a past 30 years-period (1981–2010) vs. the past 7 years (2011–2017). Climatic indices: A) PCRPTOT- annual total wet-day precipitation, B) R95PTOT- precipitation on very wet days, C) R10MM- number of heavy precipitation days, D) R20MM- number of very heavy precipitation days, E) RX1Day PCPN- monthly max 1 day precipitation amount, F) RX5Day PCPN- monthly max. 5 day precipitation amount, G) CDD- consecutive wet days, H) CWD- consecutive dry days, and I) SDII- simple daily precipitation intensity index.
(TIF)

**S4 Fig. Detailed people's perceptions on climate change and variability.** Heat map of the perceptions P1-P6 on climate change and climatic variability (% participants; p-value from the Chi-square test) in accordance with the altitudinal residence of the participants in the Lowland, Midland and Highland or their general residence in Central Nepal.
(TIF)

**S5 Fig. Socio-demographic characteristics of the study participants in Nepal.** Heat map of the socio-demographic characteristics (% participants; p-value from the Chi-square test) in accordance with the altitudinal residence of the participants in the Lowland, Midland and Highland or their general residence in Central Nepal.
(TIF)

**S6 Fig. The match between the perception P1-P2 and the annual precipitation/climate parameters.** Heat map of the people's perceptions of P1 and P2 (given as fractions of the total answers) and the trends (Sen' slope) over the last 37 years (1981–2017) of the heat or cold-related annual climate extreme indices (Table 1) in accordance with the residence of the participants in the Lowland, Midland and Highland or their general residence in Central Nepal. Significant climate indicators from the trend analyis ((Mann- Kendall test) over the last 37 years (1981–2017) are marked with an asterisk (* p < 0.05, ** p < 0.01, *** p < 0.0001). Significant climate indicators (Mann-Whitney U-test) in the 30 years (1981–2010) vs. 7 years (2011–2017) comparison after FDR correction are marked in bold. In the Highland no tropical nights (TR) were recorded.
(TIF)

**S7 Fig. The match between the perceptions P4-P6 and the annual precipitation/climate parameters.** Heat map of perceptions P4 –P6 (given as fractions of the total answers) and the trend (Sen's slope) over the last 37 years (1981–2017) of annual precipitation-related climate

extreme indices (Table 1) in accordance with the residence of participants in the Lowland, Midland and Highland or their general residence in Central Nepal. Significant indicators from the trend analyis (Mann- Kendall test) over the last 37 years (1981–2017) are marked with a asterisk(* $p < 0.05$, ** $p < 0.01$, *** $p < 0.0001$). Significant climate indicators (Mann-Whitney U-test) in the 30 years (1981–2010) vs. 7 years (2011–2017) comparison after FDR correction are marked in bold.
(TIF)

**S8 Fig. Precipitation (mm) in the last 30 years (1981–2010) vs. the last 7 years (2011–2017).** Precipitation is shown for the regions A) Lowland, B) Midland, C) Highland, and D) Central Nepal.
(TIF)

**S9 Fig. Temporal changes of the monsoon season in Nepal.** Monsoon onset (A), monsoon withdrawal (B) and the monsoon period (C; in days) from 1981–2017 in Central Nepal. Data of the year 1980 is missing. Figures are adjusted using data/figures from the Department of Hydrology and Meteorology, Kathmandu Nepal ([http://dhm.gov.np/download/](http://dhm.gov.np/download/)). The 37-year trend analysis: Sen's slope: A) -0.02 (p = not significant), B) 0.65 (p<0.0001), C) 0.71 (p<0.0001). 30 years vs. 7 years analysis: Mann-Whitney test: A) p = not significant, B) p<0.01, C) p<0.05).
(TIF)

**S1 Table. Current knowledge of climate and/or social studies on climate change impacts conducted in the Hindu Kush Himalayan region or other relevant mountains regions in the world.**
(XLSX)

**S2 Table. Description of weather stations per study region (details about altitude and distances from social data collection sites).**
(DOCX)

**S3 Table. Amount of missing values of climate extreme indices (%).** Given per study region (Lowland, Midland and Highland) and per 30-year baseline (1981–2010) or for the last 7 years (2011–2017).
(DOCX)

**S1 File. Checklist of inclusively in global research questionnaire.**
(DOCX)

**S2 File. Household survey questionnaire.**
(DOCX)

**S3 File. Focus group discussion (FGDs) and Interview guidelines.**
(PDF)

**S4 File. Detail results on climate extreme indices and social survey.**
(DOCX)

## Acknowledgments

We thank the Nepal Health Research Council for helping us to organize the survey and to provide the ethical clearance. We are grateful to the Department of Hydrology and Meteorology, Kathmandu, Nepal for providing the set of climate data and information. We thank Tamanna Neupane, Susma Baniya, Anuja Ghimire, Alisha Adhikari, Santoshi Bhandari, Diksha Parajuli,

Ichchha Thapa Magar and Sujan Thapa for their contributions during the data collection, Ute Germann, for helping with literature searches, and Gabriele Volante and Markus Braun for providing administrative support. We also express our sincere gratitude to all individuals who agreed to participate in this study.

## Author Contributions

**Conceptualization:** Ulrich Kuch, Bodo Ahrens, Meghnath Dhimal, Ruth Müller.

**Data curation:** Parbati Phuyal, Isabelle Marie Kramer, Indira Kadel.

**Formal analysis:** Parbati Phuyal, Isabelle Marie Kramer.

**Funding acquisition:** Ruth Müller.

**Methodology:** Parbati Phuyal, Isabelle Marie Kramer.

**Project administration:** Axel Magdeburg, David A. Groneberg.

**Resources:** David A. Groneberg, Meghnath Dhimal, Ruth Müller.

**Software:** Axel Magdeburg.

**Supervision:** Edwin Wouters, Ulrich Kuch, Bodo Ahrens, Mandira Lamichhane Dhimal, Meghnath Dhimal, Ruth Müller.

**Validation:** Parbati Phuyal, Isabelle Marie Kramer, Bodo Ahrens, Mandira Lamichhane Dhimal, Meghnath Dhimal, Ruth Müller.

**Visualization:** Parbati Phuyal, Isabelle Marie Kramer, Ruth Müller.

**Writing – original draft:** Parbati Phuyal, Isabelle Marie Kramer.

**Writing – review & editing:** Indira Kadel, Edwin Wouters, Axel Magdeburg, David A. Groneberg, Ulrich Kuch, Bodo Ahrens, Mandira Lamichhane Dhimal, Meghnath Dhimal, Ruth Müller.

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
