## [Decision Letter · Decision Letter 0]

13 Nov 2024

PONE-D-24-46221On people’s perceptions of climate change and its impacts in a hotspot of global warmingPLOS ONE

Dear Dr. Phuyal,

Thank you for submitting your manuscript to PLOS ONE. After careful consideration, we feel that it has merit but does not fully meet PLOS ONE’s publication criteria as it currently stands. Therefore, we invite you to submit a revised version of the manuscript that addresses the points raised during the review process.

Based on the review input of external reviews, the reviewer - 2 has suggested for some minor revisions or revisit manuscript for possible answer the queries.

We look forward to receiving your revised manuscript.

Kind regards,

Tasawar Baig, PhD

Academic Editor

PLOS ONE

“The work was funded by the Federal Ministry of Education and Research of Germany (BMBF) under the project AECO (Number 01Kl1717) as part of the National Research Network on Zoonotic Infectious Diseases of Germany.”

Reviewers' comments:

Reviewer's Responses to Questions

**Comments to the Author**

1. Is the manuscript technically sound, and do the data support the conclusions?

Reviewer #1: Yes

Reviewer #2: Yes

2. Has the statistical analysis been performed appropriately and rigorously? 

Reviewer #1: Yes

Reviewer #2: Yes

3. Have the authors made all data underlying the findings in their manuscript fully available?

Reviewer #1: Yes

Reviewer #2: Yes

4. Is the manuscript presented in an intelligible fashion and written in standard English?

Reviewer #1: Yes

Reviewer #2: Yes

5. Review Comments to the Author

Reviewer #1: Review Comments on "On People’s Perceptions of Climate Change and Its Impacts in a Hotspot of Global"

• The paper presents valuable insights into people's perceptions of climate change in a critical area. The introduction effectively sets the context and highlights the importance of understanding local perspectives.

• Overall, the abstract is well-written. However, it would be beneficial to mention the different altitudinal regions covered in the study. Including a line on policy recommendations or how this study can inform climate change adaptation strategies for mountain communities would enhance its impact.

• The introduction is generally well-structured. It could be strengthened by integrating some reports from ICIMOD, which have conducted extensive work related to climate change and disaster risk assessment in Nepal.

• The methodology is generally well-organized, but further clarification is needed on the following points:

o Sample Selection: More detail on how participants were chosen would enhance the study's credibility.

o Data Collection Methods: Elaborating on the tools used (e.g., surveys, interviews) and any validation processes undertaken would be beneficial.

o Statistical Analysis: A more thorough explanation of the analytical techniques used would improve the transparency of the results.

o Please add a few lines regarding household consent and ethical approval processes.

• Including a map of the study area and the geographic locations of data collection would be a valuable addition.

• Clarify the consent obtained from the relevant department for data usage. Specifically, detail the procedure for downloading data from their website (e.g., the withdrawal of the monsoon and monsoon season length obtained from the Department of Hydrology and Meteorology, Kathmandu, Nepal).

• The results are clearly presented. However, consider providing more contextual analysis that links the results to existing literature, which would strengthen the argument and illustrate how findings align or contrast with previous studies.

• Some discussion points could benefit from deeper exploration, particularly concerning the implications of the findings on local policy and community actions. It would be helpful to mention whether climate-induced disaster risk has increased or decreased and how community perceptions relate to crop production changes.

• If possible, please provide the statistical values (p-values) regarding community perceptions of climate change, as mentioned in the data. It would also be useful to present p-values for lowland, midland, and highland areas in a table titled "People's Perceptions on Climate Change."

• The conclusion summarizes the main findings well, but it would be beneficial to include more specific recommendations for stakeholders based on the study's findings. Additionally, highlighting the limitations of the study and suggesting areas for future research would provide a more balanced view.

• References: Ensure all references are up-to-date and relevant, particularly in the context of climate change literature. This will enhance the paper's credibility and scholarly contribution.

Reviewer #2: The paper offers valuable insights into peoples’ perceptions of climate change in selected sites of Nepal systematically compares those perceptions with climate indicators derived from meteorological data. Hence the study investigates specific socio-economic and other contexts (differed across the altitudinal regions) shaping peoples’ perceptions and provide useful recommendation for adaption and mitigation measures. Please see the attached file for the specific comments, and suggestions.

6. PLOS authors have the option to publish the peer review history of their article (what does this mean?). If published, this will include your full peer review and any attached files.

Reviewer #1: No

Reviewer #2: **Yes: **Muhammad Zafar Khan

---

## [Author Response · Author response to Decision Letter 0]

20 Dec 2024

Dear Editor and dear Reviewers, 

We have carefully revised the manuscript as per the suggestions of Editor and the reviewers. In the current submission, we have added Fig 1 (Map of study area), therefore the previous figures: Fig 1, Fig 2, Fig 3 and Fig 4 has now renamed as Fig 2, Fig3, Fig 4 and Fig 5 respectively. We have uploaded the household questionnaire and interview guidelines in the reviewed submission as S2 and S3 files. Therefore, the previous file S1 has now renamed as S4 files. Thank you. All other changes are reported in the rebuttal letter. Thank you.

Sincerely yours 

Parbati Phuyal

---

## [Editor Report · Decision Letter 1]

5 Jan 2025

On people’s perceptions of climate change and its impacts in a hotspot of global warming

PONE-D-24-46221R1

Dear Parbati Phuyal,

We’re pleased to inform you that your manuscript has been judged scientifically suitable for publication and will be formally accepted for publication once it meets all outstanding technical requirements.

Kind regards,

Tasawar Baig, PhD

Academic Editor

PLOS ONE
---

## [Editor Report · Acceptance letter]

14 Jan 2025

PONE-D-24-46221R1 

PLOS ONE

Dear Dr. Phuyal, 

I'm pleased to inform you that your manuscript has been deemed suitable for publication in PLOS ONE. Congratulations! Your manuscript is now being handed over to our production team.

Kind regards, 

on behalf of

Professor Tasawar Baig 

Academic Editor

PLOS ONE